# Biomaterials and Encapsulation Techniques for Probiotics: Current Status and Future Prospects in Biomedical Applications

**DOI:** 10.3390/nano13152185

**Published:** 2023-07-27

**Authors:** Qiqi Sun, Sheng Yin, Yingxu He, Yi Cao, Chunping Jiang

**Affiliations:** 1Jinan Microecological Biomedicine Shandong Laboratory, Shounuo City Light West Block, Jinan 250117, China; sunqiqi971018@163.com (Q.S.); yuhuashi.1209@163.com (S.Y.); 2Collaborative Innovation Center of Advanced Microstructures, Nanjing University, Nanjing 210093, China; 3School of Computing, National University of Singapore, Singapore 119077, Singapore; yingxu.he1998@gmail.com; 4Department of Hepatobiliary Surgery, Affiliated Drum Tower Hospital, Medical School of Nanjing University, Nanjing 210000, China; 5State Key Laboratory of Pharmaceutical Biotechnology, Jiangsu Key Laboratory of Molecular Medicine, Medical School of Nanjing University, Nanjing 210000, China

**Keywords:** probiotics, tissue engineering, biomimetic materials, microecology

## Abstract

Probiotics have garnered significant attention in recent years due to their potential advantages in diverse biomedical applications, such as acting as antimicrobial agents, aiding in tissue repair, and treating diseases. These live bacteria must exist in appropriate quantities and precise locations to exert beneficial effects. However, their viability and activity can be significantly impacted by the surrounding tissue, posing a challenge to maintain their stability in the target location for an extended duration. To counter this, researchers have formulated various strategies that enhance the activity and stability of probiotics by encapsulating them within biomaterials. This approach enables site-specific release, overcoming technical impediments encountered during the processing and application of probiotics. A range of materials can be utilized for encapsulating probiotics, and several methods can be employed for this encapsulation process. This article reviews the recent advancements in probiotics encapsulated within biomaterials, examining the materials, methods, and effects of encapsulation. It also provides an overview of the hurdles faced by currently available biomaterial-based probiotic capsules and suggests potential future research directions in this field. Despite the progress achieved to date, numerous challenges persist, such as the necessity for developing efficient, reproducible encapsulation methods that maintain the viability and activity of probiotics. Furthermore, there is a need to design more robust and targeted delivery vehicles.

## 1. Introduction

Probiotics are beneficial living microorganisms commonly used to improve gut microbiota balance [1]. *Lactobacillus* and *Bifidobacterium* are common probiotic strains often used in products, displaying strong anti-inflammatory properties [2,3]. In addition, probiotics provide other health benefits, such as improving digestion, boosting the immune system, and fighting viruses [4]. *Lactic acid bacteria*, the main type of probiotics, can modulate the human gut microbiota by suppressing the growth of opportunistic bacteria. The potential health claims related to probiotics are numerous, ranging from maintaining a healthy intestinal flora and protecting against infections to alleviating lactose intolerance and stimulating the immune system [5]. Clinical trials have demonstrated that probiotics can improve outcomes in immune-system-related conditions, viral infections, atopic dermatitis, rheumatoid arthritis, allergic conditions, and respiratory tract infections [6,7].

As our understanding of the benefits of probiotics and gut microorganisms increases, the demand for probiotics continues to grow. Currently, probiotic foods account for 70% of the functional food market [8]. In fact, the global probiotics market was valued at $4.62 billion in 2019, and it is expected to reach $7.59 billion by 2026 [9].

However, the performance and survival rate of probiotics are influenced by various complex factors, such as the presence of acids and bile, specific ions, nutrient depletion, osmotic pressure, oxidative stress, and the passage through the gastrointestinal tract [10,11,12]. To improve the effectiveness and efficiency of probiotics, researchers are exploring the use of polymers in probiotic encapsulation technology. This technology involves techniques such as microencapsulation, which is the process of enclosing probiotics within a substrate to shield them from harsh environmental conditions and create an optimal microenvironment for their survival and stability [13,14]. Another approach is to integrate probiotics into electrospun fibers, which helps stabilize and activate the probiotic cells. Electrospun fibers also provide a larger surface-area-to-volume ratio, allowing for rapid dissolution or controlled release of probiotics [15,16]. Encapsulating probiotics within polymers offers several advantages. It provides a protective shield around the bacteria, safeguarding them while allowing the passage of small molecules [17]. Additionally, encapsulation maintains probiotic stability and allows for a higher probiotic load [18,19]. It also enables controlled and prolonged release of probiotics, supports their reproduction, and promotes adherence and extended residence time within the body [20,21,22]. In addition to encapsulation technology, there are also various material options for encapsulating probiotics. As an effective encapsulation material, the selected biomaterial must be able to protect the encapsulated probiotics and allow them to reach the targeted site (small intestine/large intestine) along the gastrointestinal tract, where they can exert their health-promoting effects [23]. The release of encapsulated probiotics from the encapsulation material occurs only when certain environmental conditions, such as temperature, pH, and enzymatic activity, are met [24]. This article will elaborate on several common probiotic packaging materials. 

Overall, studies have shown that the combination of probiotics with polymers can enhance their safety and effectiveness [25,26]. Nevertheless, the progress in developing effective techniques for encapsulating probiotics remains restricted [27]. Despite efforts, there is currently no capsule system available that can guarantee the preservation of all probiotics’ activities. This review aims to elucidate and integrate the advantages and disadvantages of various encapsulation techniques, materials, evaluation methods, etc., for probiotics, with the goal of promoting the advancement in this field. Furthermore, it underscores the commercial difficulties associated with probiotic encapsulation and offers a glimpse into potential future developments in this domain.

## 2. The Necessity of Encapsulating Probiotics with Polymers

To ensure optimal performance of these beneficial bacteria in carrying out their biological functions, it is necessary to have a minimum of 10^7^ CFU/mL or gram of the product being used [28]. However, the production, processing, storage, and transportation of probiotics can lead to their inactivation. Various environmental factors, such as temperature, pH, and light, can affect the viability of probiotics [29,30]. During transportation and storage, probiotics can be exposed to high temperatures, vibration, and humidity, which can result in their inactivation. High oxygen levels can also negatively impact the viability of probiotics, as they are primarily microaerophilic or anaerobic [31]. Moisture and humidity can activate bacteria prematurely, leading to degradation. Additionally, when probiotics are stored alone, they lose the support and protection of symbiotic microorganisms, which can affect their growth and survival [32]. To mitigate these challenges and ensure the survival and effectiveness of probiotics, protective carrier materials such as polymer packaging can be used during transportation and storage. Polymer encapsulation can maintain the stability of probiotics, extend their lifespan, and protect them from external factors such as pressure, temperature changes, and oxidation [33,34].

In addition to external factors, the internal environment of the human body can also contribute to the inactivation of probiotics [35]. Many types of probiotics can be negatively impacted by highly acidic gastric fluids, which typically have a pH ranging from 1 to 3 [36]. A study conducted earlier examined the viability of various commercially available probiotics during their journey through the gastrointestinal tract (GIT). To the surprise of many, the findings highlighted that within just five minutes of incubation in gastric fluids, every tested commercial product experienced a considerable 10^6^-fold reduction in colony-forming units (CFU) [37]. In the small intestine, upon arrival at the colon, probiotics encounter the challenge of competing with preexisting bacteria, attaching to the intestinal mucus layer, and subsequently thriving and reproducing. Digestive enzymes and biles reduce the adhesion of strains [38,39]. Figure 1 shows the factors that may cause probiotics to be inactivated. Consequently, there is concern that many commercial probiotic products may be ineffective due to the challenges they face during food processing, storage, and passage through the upper gastrointestinal tract. Even if they reach the colon, there is no guarantee that they will establish themselves as part of the gut microbiome and may be excreted in feces [37,40]. Polymer-encapsulated probiotics can protect probiotics from external factors such as pressure, temperature changes, oxidation, etc., during food processing and storage [41,42]. For example, encapsulated probiotics can be stored better in milk [43]. In addition, polymer encapsulation can also help probiotics survive and function better in the human intestinal tract [44]. Therefore, polymer encapsulation is an effective method to ensure the survival rate of probiotics and improve their health benefits.

## 3. Polymer Choice for Combining Probiotics

In the biomedical field, biopolymer materials combined with probiotics are also favored due to their good biocompatibility and regulability. Biopolymer materials are a class of biological materials that can be used in the fields of cell tissue engineering, drug delivery and biosensors [45]. Biopolymer materials combined with probiotics realize the carrier function while maintaining the activity of probiotics. These materials are generally required to be nontoxic and to achieve targeted release [46]. At present, several materials have been investigated for use in tissue engineering in combination with probiotics; among these, polysaccharides and protein materials have emerged as the most commonly used materials for embedding probiotics. The utilization of various technologies encompassing food-grade polymers such as gelatin, alginate, chitosan, carrageenan, pectin, and carboxymethyl cellulose (CMC) is quite prevalent and has been explicated in Table 1.

### 3.1. Encapsulation in Polysaccharides

Polysaccharides have the advantages of high stability, low immunogenicity, and abundant availability, but they also have reactive functional groups, sensitivity to moisture, and brittleness [47,48]. 

Alginate (Alg) hydrogels are widely recognized as a cost-effective and versatile method for encapsulating probiotics. They consist of two monosaccharide units, D-mannuronic acid (M) and α-L-guluronic acid (G), linked together in a 1–4 configuration [49]. Alg is insoluble in acidic conditions, which can provide protection for probiotics in acidic environments [50]. The carboxylic group of Alg can cross-link with divalent cations to form a hydrogel [51]. It is worth noting that alginate microcapsules not only have advantages in improving the stability, survival rate, and targeting of probiotics but also have simple, fast, and inexpensive production [52,53]. Vega-Carranza et al. utilized ionic gelation to encapsulate *B. licheniformis* BCR 4–3 marine probiotics in alginate particles (AMPs) to improve storage stability and achieve targeted delivery within shrimp intestines. The results indicate that AMPs are a promising method for delivering these probiotic bacteria while maintaining their effectiveness and stability [54]. Henk J. Busscher et al. compared the viability and resistance to gastric acid and tetracycline of Bifidobacterium bifidum loaded on different carriers in various experimental groups. The comparative study showed that only the alginate hydrogel shell provided protection against simulated gastric acid and tetracycline. The probiotic loaded in alginate hydrogel showed a synergistic effect with tetracycline, which could kill tetracycline-resistant *E. coli* and maintain the integrity of the intestinal epithelial cell layer and its barrier. The synergistic effect between the alginate hydrogel and antibiotic when loading probiotic Bifidobacterium bifidum is worthy of further research to treat antibiotic-resistant *E. coli* gastrointestinal infections [53]. However, the porous structure and hydrophilic properties of Alg hydrogels do not provide sufficient protection to maintain probiotic viability in the stomach, and monovalent ions can further destabilize the structure [55]. An effective solution to this issue is the addition of a cationic coating, which can reduce pore size and maintain probiotic cytoplasmic pH, promoting probiotic survival [56]. Microencapsulation using calcium alginate and other alginates is one of the most commonly used methods for embedding probiotics due to its low cost, ease of use, and good biocompatibility. Ana Jaklenec et al. selected a commercial composite probiotic preparation and used the calcium alginate microencapsulation technique to embed the probiotics. The findings of their study indicate that the calcium alginate microcapsule functions similarly to a biological membrane, enhancing the probiotics’ resistance to antibiotics without affecting their own metabolism [57]. The low survival rate of probiotics loaded in calcium alginate hydrogels by freeze-drying has limited their commercial application. According to the findings of Zhong Fang et al., a comparison of the effects of calcium alginate and sodium alginate on the activity loss of encapsulated probiotics during freeze-drying revealed that calcium caused damage to both the cell wall/membrane and intracellular homeostasis. Furthermore, *Lactobacillus rhamnosus GG* (*LGG*) encapsulated in sodium alginate exhibited a higher survival rate than that encapsulated in calcium alginate [58]. The coalescence of alginates with other biopolymers in the production of hydrogels can enhance the encapsulation capabilities and viability of probiotics in comparison with using alginates alone. Qian Chen et al. used effective biofilms such as chitosan and sodium alginate to encapsulate *Escherichia coli* Nissle 1917 (ECN) via a layer-by-layer electrostatic self-assembly strategy. This approach increased the abundance of intestinal microorganisms that maintain intestinal homeostasis, laying the foundation for the development of therapeutic proteins for the treatment of intestinal-related diseases using probiotics [50].

Chitosan (CS) is a cationic polysaccharide consisting of D-glucosamine and N-acetyl-glucosamine residues linked by β-(1 → 4) bonds. It is commercially derived by partially deacetylating chitin extracted from crustaceans [59]. In addition, chitosan forms a gel structure through ionotropic gelation and is soluble in pH < 6, similarly to alginate. However, the particle size of chitosan-coated alginate beads is relatively large due to additional alginate coating or some aggregation of microgels [60]. Its important characteristics include its unique polymeric cationic nature, biocompatibility, non-toxicity, and biodegradability [61]. CS can present disadvantages as an encapsulating material for probiotic bacteria due to its inhibitory effect on microorganisms, including lactic acid bacteria. Therefore, CS is commonly utilized as a coating or shell rather than a capsule [62]. Studies have shown that using CS as a coating material in alginate beads can improve the survival of encapsulated bacteria in harsh conditions like the gastrointestinal tract and high temperatures. Encapsulating different strains of bacteria can present different behaviors [63]. Coating alginate beads with chitosan has been found to create a complexation between the two materials, resulting in important properties such as reduced porosity, decreased encapsulated bacteria leakage, and high stability across varying pH ranges. This is due to the negative charge of alginate interacting with the positive charge of chitosan, which forms a semi-permeable membrane. The resulting capsules have a smoother surface and lower permeability to water-soluble molecules [64]. For example, CS-coated alginate microcapsules were found to be the most effective technology for protecting probiotic bacteria (such as *Lactobacillus* and *Bifidobacterium* spp.) against all tested conditions [65]. Shuyu Xi et al. showed that the properties of CS nanoparticles prepared by ionotropic gelation could be controlled consistently by adjusting the formula parameters. The optimal average particle size of chitosan nanoparticles was 67.60 ± 0.11 nm, the zeta potential was +33.23 ± 1.20 mV, and the aggregation index was 0.26 ± 0.00 [66]. Additionally, CS can be utilized for encapsulating probiotics through methods such as thermally induced phase separation, freeze gelation, and photo crosslinking. These processes do not involve the use of toxic crosslinking agents, providing us with multiple options [67,68]. Another study indicated that microencapsulation of *L. gasseri* and *B. bifidum* using quercetin as prebiotics and chitosan as coating material in alginate microparticles resulted in enhanced survival rates during simulated gastrointestinal conditions [69]. B. Lindman et al. used hybrid particles of carboxymethyl cellulose–CS to encapsulate *LGG*. The results showed successful encapsulation of the probiotics, and the particles were stable at gastric pH and significantly swelled at intestinal pH. This method can regulate the gut microbiota and improve human health [70].

Although not as commonly employed as the aforementioned materials for probiotic encapsulation, several other types of polysaccharides exhibit promising potential as matrices for protecting encapsulated bacteria from the unfavorable acidic and bile conditions of the stomach. Mengzhou Zhou et al. conducted a study using polysaccharides from *C. axillaris* peels (CP) as a polymer filler to decrease the permeability of alginate microgels and increase their protective effect on probiotics. CP has excellent emulsifying activity and can encapsulate lipid droplets. As emulsions can fill the pores in the hydrogel matrix, thus slowing down the diffusion process, adding emulsions to the biopolymer microgel can also enhance the viability of probiotics during storage and passage through the gastrointestinal tract [71].

In the field of probiotic encapsulation, polysaccharides possess several characteristics, including: (1) Ion-induced gelation: Polysaccharides can form cross-linked hydrogel structures through interactions with specific ions [72]. (2) Structural reinforcement: Polysaccharides resist enzymatic degradation and withstand acidic environments, enhancing their stability [73]. (3) Enteric dissolution: Polysaccharides dissolve only in the intestinal environment, ensuring targeted delivery [74]. (4) Charge interaction: Polysaccharides can interact with other polysaccharides or proteins with opposite charges [75]. (5) Prebiotic properties: Polysaccharides can be selectively utilized by host microorganisms, providing health benefits [76]. However, traditional polysaccharides often struggle to meet all of these requirements simultaneously. Therefore, researchers are interested in exploring novel polysaccharide materials for use as coating materials in probiotic encapsulation. These new polysaccharides include modified celluloses such as methylcellulose (MC), hydroxypropyl methylcellulose (HPMC), and carboxymethyl cellulose (CMC), among others [77,78,79].

Composite hydrogels, using materials such as carboxymethyl cellulose, xanthan gum, locust bean gum, nanocellulose, and clay have shown potential in enhancing the gastric survival of encapsulated probiotics. However, the increased viscosity of the mixed slurry may pose challenges during manufacturing, particularly in the extrusion process. Additionally, composite hydrogels are prone to strong molecular interactions, which could delay their disintegration and release of encapsulated probiotics in the host intestine [55]. Therefore, researchers are attempting to use different techniques to prepare materials to improve these limitations. Another issue that needs to be noted is that a larger number of layers in the material that envelops the probiotics is not necessarily better. Dimitris Charalampopoulos et al. used chitosan-sodium alginate capsules to encapsulate probiotics. After one layer of capsules was treated with simulated gastric fluid, the viability of probiotics was 83.4% (compared with 32.5% for nonencapsulated probiotics). When the number of chitosan layers increased to 2 and 3, the viability of probiotics also increased to 89.3% and 95.8%, respectively. As the thickness of the chitosan coating increased, the probiotic survival rate decreased. This could be attributed to the enlargement of capsules and decreased cross-linking density, which led to more digestive fluids breaking down the probiotics [80]. Although increasing the number of coating layers could enhance the survival of probiotics in surface-coated hydrogels, this process is time-consuming and could lead to adverse effects due to coating materials during storage [51,81]. Therefore, it is necessary to control the number of layers in the material that envelops the probiotics.

### 3.2. Encapsulation in Protein

The benefits of protein encapsulation include nutritional value, and lower allergenicity, but proteins have an undesirable flavor and low digestibility [82,83,84]. Gelatin, whey protein, and casein are commonly used proteins for the encapsulation of probiotics. This is due to their amphiphilic nature, which makes them suitable for the task.

In the encapsulation of probiotic bacteria, gelatin is frequently utilized as a coating material, either on its own or in conjunction with other supporting materials. Gelatin is composed of 18 different complex amino acids, of which approximately 57% are glycine, proline, and hydroxyproline. The remaining approximately 43% consists of other notable amino acid families, such as glutamic acid, alanine, arginine, and aspartic acid [85]. It is a thermoreversible gelling material and its temperature-dependent nature enables its properties to be modified when subjected to different temperatures [86]. Gelatin is known to possess both anionic and cationic properties and can be combined with other anionic polysaccharides such as gellan gum. Gelatin and gellan gum are capable of developing a consistent blend when pH exceeds 6, owing to their shared negative charges. When the pH of gelatin solution decreases below its isoelectric point, the solution gains positive charges which facilitate its interaction with gellan gum [87,88]. The gelatin-toluene blend has the ability to produce more robust capsules compared with using just the gelatin solution by itself. At higher concentrations, these capsules exhibit greater resistance to mechanical stress, such as cracking [89]. Li Lanjuan et al. found that encapsulating Lactobacillus paracasei Li05 in alginate–gelatin microgel and buffering with MgO significantly enhanced the probiotic’s activity and stability. The probiotic bacteria loaded with MgO microgel were more stable than free bacteria or probiotic bacteria in separate microgel [90]. Polysaccharides and protein materials can also be combined for encapsulation of probiotics. Subrota Hati et al. used tea protein/xanthan gum (TP/XG) to encapsulate probiotics, which significantly improved their survival ability under heat treatment and simulated gastrointestinal conditions [91].

Whey proteins (WP) are favored biomaterials for encapsulating probiotics due to their positive attributes. They have been discovered to enhance the resilience of probiotics [92]. Various whey protein products with different protein contents, ranging from whey powder (approximately 15% protein) to whey protein isolates, have been employed in the microencapsulation (ME) process of probiotics [93,94]. Researchers have found that the inherent nature of whey proteins plays a crucial role in capturing the probiotic strain *L. rhamnosus GG*. The characteristics of the food matrix containing probiotics significantly affect their viability. Consequently, it has been established that the type of whey proteins utilized can influence the efficient entrapment of *L. rhamnosus GG* [95].

### 3.3. Encapsulation in Lipids

Lipids increase the efficiency of probiotics, can form nanocapsules with low energy input, and are suitable for different active agents, but they can undergo oxidation, leakage, and fusion, and are thermally unstable above a certain temperature [96,97]. Lipids are a group of naturally occurring molecules divided into various subcategories such as fats, mono-, di-, and triglycerides, phospholipids, sterols, and waxes [98]. Due to their high acceptance, low toxicity, and adaptability for use in food products, lipid-based delivery methods are gaining attention among food scientists as a means to deliver bioactive components in food systems. Two potential lipid-based delivery systems, namely liposphere (solid lipid particles (SLP)) and liposome, are currently utilized for encapsulating probiotics in food and pharmaceutical applications [99]. Liposomes are spherical lipid vesicles that typically range in size from 50 to 500 nm in diameter. They are made up of one or more lipid bilayers, which are formed by emulsifying natural or synthetic lipids in an aqueous medium [100]. Liposomes have advantages such as suitability for encapsulating hydrophobic ingredients, sustained release effect, and ability to be produced in huge quantities commercially [101,102]. But they are not suitable for food applications due to high content of saturated fatty acids and limited capacity to hold hydrophobic compounds in their core [96,103]. Lipid droplets are comprised of a central core consisting of neutral lipids, surrounded by a phospholipid monolayer that contains embedded or loosely associated proteins. Their advantages include suitability for various bioactive ingredients, commonly used as encapsulation carriers. Similarly to the cell membrane structure, they facilitate transport of bioactive substances and protects them from digestion, and can be commercially produced on a large scale [104,105]. However, thermal sensitivity limits applications in high temperature processes [106]. Nuria C. Acevedo et al. demonstrated that lipids can protect probiotics from the harsh conditions of the digestive tract [107].

### 3.4. Encapsulation in Synthetic Polymers

Synthetic polymers, including polyesters like poly (D,L-lactic-co-glycolic acid) (PLGA), polyacrylamides, and polyvinyl alcohol (PVA), have also been explored for delivering probiotics. Zhang Xianzheng et al. developed a formulation of live bacteria (Hy@Rm) by combining skin-symbiotic bacteria from *Roseomonas mucosa* with polyvinylpyrrolidone (PVP), PVA, and sodium alginate in a skin dressing using Ca^2+^-mediated cross-linking and freeze-thaw (F-T) cycling. This skin dressing can downregulate pro-inflammatory factors, accelerate epithelial cell regeneration and wound healing, and alleviate atopic dermatitis [108].

However, the use of synthetic polymers like PLGA is restricted for probiotic encapsulation due to the involvement of organic solvents that are needed to dissolve the polymers during the fabrication process, which can potentially harm the probiotics.

**Table 1 nanomaterials-13-02185-t001:** Different polymers that encapsulate probiotics.

Reference	Bacteria Systems	Polymers	Functionality
Hong, X. et al. [109]	*Lactobacillus acidophilus*	Polygamma-glutamic acid hydrogel	The microcapsule is NO responsive, rapidly releases probiotics, maintains the intestinal mechanical barrier, and regulates the intestinal flora balance.
Deng, J. et al. [110]	*Lactococcus lactis*	Heparin grafted with Poloxam 407	Protect the release of probiotic VEGF without loss of activity, and can limit the spread of VEGF to achieve local release.
Liu, W. et al. [111]	*Lactobacillus reuteri*	Methylacrylylated gelatin (GelMA), methylacrylylated hyaluronic acid (HAMA)	Protecting probiotics from the attack of the immune system; also helps prevent the potential threat of probiotics escaping.
Liu, J. et al. [112]	*E. coli* Nissle 1917	Yeast membrane cell coating (YMs)	The survival rate of gastric acid and bile salts was improved, and the β-glucan contained in yeast membrane was recognized by the Dectin-1 receptor of M cells and promoted the uptake of M cells, thus stimulating a strong mucosal immune response.
Neel S. Joshi. et al. [113]	*E. coli* Nissle 1917	Fusion protein hydrogel	Genetically programmed *E. coli* can secrete curly fusion TFFs without pathogenicity. It enhanced the protective effect on mice colitis induced by sodium dextran sulfate and was related to mucosal healing and immune regulation.
Zeng, R. et al. [114]	*Lactobacillus plantarum*	Oxidized Bletilla Polysaccharide (OBSP), Chitosan (CS)	Maintain wound moisture, promote VEGF factor high expression, inhibit inflammation, accelerate collagen deposition, prevent scar formation, and significantly promote wound healing.
Zhou, Q. et al. [115]	*Lactobacillus rhamnosus*	Dihydrazine adipate Modified hyaluronic acid (HA-ADH), PF127-CHO, polysaccharide fucosan sulfate (FD)	Has enhanced antibacterial properties and promotes superbug-induced wound healing.
Duan, J. et al. [116]	*Lactobacillus rhamnosus*	Sulfhydrylated hyaluronic acid (HA-SH)	Protect probiotics from gastrointestinal acid, bile acid, and other stresses.
Shrivastava, S. et al. [117]	*Lactobacillus acidophilus* and *Lactobacillus casei*	glucose, galactose, rhamnose, galactouronic acid and fucose	Demonstrated protection against simulated gastrointestinal conditions at high and low temperatures, ensuring higher viability of embedded probiotic cells.
Jin, W. et al. [118]	*Lactobacillus rhamnosus*	Bentonite, alginate	The survival rate of *LGG* under gastric pH value is improved. Complete intestinal release of *LGG* was observed after hydrogel decomposition.
Zhou, L. et al. [119]	*Bifidobacterium lactis*	Whey protein isolate (WPI), pectin, D-gluconate-delta-lactone (GDL), Calcium	Increased probiotic activity, especially after exposure to the stomach stage.
Lou, Y. et al. [120]	*Lactobacillus paracei*	Starch, metal ions	High gastric acid tolerance, showing excellent controlled release properties against probiotics

## 4. Techniques and Evaluations of Probiotic Encapsulation

### 4.1. Probiotic Encapsulation Technology

Encapsulation technology for probiotics has been used as a carrier or targeted delivery system, including freeze-drying, emulsification, and extrusion, which have been shown to have the potential to enhance the survival and biological activity of probiotics [121]. Although the use of high temperatures or organic solvents limits their use, researchers have optimized the encapsulation process by improving technical parameters and using additives [122,123]. When incorporating encapsulated probiotics into a food product, it is crucial that the capsules are capable of withstanding both the production and storage processes, as well as the digestive system. However, most scientific studies only focus on improving one aspect without considering the other. Probiotic encapsulation technology can be divided into microencapsulation technology and electrostatic spinning technology, as shown in Figure 2. This article will introduce them in order.

#### 4.1.1. Microencapsulation

Microencapsulation refers to the technique of packaging gaseous, liquid, or solid materials inside miniature sealed capsules that can release their contents at a controlled rate under specific conditions [124]. To be efficient, a microencapsulation system needs to keep the probiotics stable during storage, shield them from the challenging conditions in the upper gastrointestinal tract, release them in the colon, and finally promote their capability to colonize the mucosal surfaces [125]. To ensure the safety and efficacy of probiotics, the microencapsulation process must provide sufficient protection against acidity and avoid damage to the bacteria [126]. Covalently or ionically cross-linked polymer networks are commonly used as microencapsulation matrices, but uncrosslinked polymer granules, produced through spray drying, can also be used. Some studies have shown that probiotic-containing tablets may enhance bacterial survival in the gastrointestinal tract [30,127]. Furthermore, targeted release across the small and large intestine can be achieved through microencapsulation. These microcapsules have a thin, spherical, strong, semipermeable membrane encapsulating a liquid or solid core that can vary in diameter from a few microns to 1 mm [128].

Current microencapsulation technologies include layer-by-layer techniques, spray drying, emulsification, extrusion, and electrospraying [129]. Table 2 provides a brief overview of the different microencapsulation methods used for microbial encapsulation of probiotics and their advantages and disadvantages. Microencapsulation has been proposed as an effective means of protecting probiotics from degradation. First, they could be designed to create a physical barrier that shields the probiotics from any harmful elements in their environment, such as stomach acid, bile salts, or digestive enzymes [126,130]. Second, they could be formulated with additives that create favorable conditions for acid-resistant probiotics, control pH levels, and promote their proliferation [131]. Finally, microparticles may be designed to capture specific compounds that enhance probiotic survival, which are released by the probiotics [132]. Microencapsulation finds its applicability in multiple facets of the food industry, including stabilizing the core material, managing oxidative reactions, providing controlled release both in terms of temporal and time-controlled mechanisms, masking flavors, colors or odors, prolonging the shelf life and safeguarding constituents against nutritional degradation [133,134,135]. Choosing a technique to trap bacteria depends on several factors, such as the possibility of producing it in large quantities, expenses, and particle shape and durability, but the most critical aspect is the achievable viable bacterial count.

Freeze-drying (FD) is widely used in the food and pharmaceutical industries to dry fragile components, as it preserves a significant amount of live probiotics [136,137]. The harm caused by low temperature and water sublimation during the freeze-drying process to microorganisms is relatively small compared with the high temperature used in spray drying, which improves the survival ability of bacteria [14]. However, this process has limitations such as being time-consuming and costly. The versatility and adjustability of layer-by-layer (LbL) self-assembly technology make it a promising method for probiotic encapsulation [138]. This technique allows for the fabrication of diverse multilayer shells with controlled compositions and structures. Ashok M. Raichur et al. encapsulated the probiotic *Lactobacillus acidophilus* using layer-by-layer self-assembly of polyelectrolytes such as polyelectrolyte (PE), chitosan (CHI), and carboxymethylcellulose (CMC). Compared with the nonencapsulated cells, nearly all the free probiotics died, while the encapsulated probiotics showed an increased survival rate of 33% in simulated gastric (SGF) and intestinal fluids (SIF) [139]. The number of layers of encapsulation needs to be taken into account when performing this approach. Probiotics can be encapsulated in microbeads made of various materials. For example, alginate beads can be formed by extrusion and emulsification to encapsulate probiotics [140,141]. These microbeads can release their contents at a controlled rate. However, in practical industrial applications, the encapsulating material and the particle diameter size of these microparticles also need to be taken into consideration.

**Table 2 nanomaterials-13-02185-t002:** Different microencapsulation methods and features employed for the encapsulation of probiotic microorganisms.

Reference	Methods	Feature
Ana Jaklenec. et al. [142]	layer-by-layer	This technology has high controllability and adjustability.
R Paul Ross. et al. [143]	spray drying	The particle size can be controlled, the cost is low, the production yield is high, and it has strong water retention ability, suitable for industrial applications. But the viability loss of the probiotics is very high and product stability is poor.
Xiaojun Ma. et al. [144]	emulsification	The production yield is high, easy to scale up, suitable for industrialization and the particle size is smaller, but there may be residual oil and the droplet size distribution is not uniform.
Amparo Lopez-Rubio. et al., [145]	electrospraying	Convenient and fast, economical and efficient, mild conditions, strong adaptability and easy to scale up.
Wee Sim Choo. et al. [146]	extrusion	The method is characterized by low cost, simple operation, mild conditions, and uniform size, but the production yield is small and the particle size is larger, difficult to use in large scale productions.
Siddalingaiya Gurudutt Prapulla. et al. [147]	freeze-drying	The product stability is good, suitable for embedding thermosensitive materials, but it is expensive, has complicated operation, and the surface of the product may wrinkle and shrink.
Costas G. Biliaderis. et al. [148]	complex coacervation	The production volume is large, but the process is complex and the cost is high.

#### 4.1.2. Electrostatic Spinning

Electrospinning is a versatile technology used to continuously produce nanofibers ranging in diameter from nanometers to micrometers [149]. First introduced by Formhals in 1934, electrospun fibers have been widely used in tissue engineering, energy storage and conversion, food packaging, drug delivery and release, catalysis, sensors, filtration, and almost all fields of research [150]. Electrospun nanofibers are mainly used for encapsulation (antioxidants, antimicrobial agents, enzymes, and probiotics) and packaging in the food industry [151].

Polyvinyl alcohol (PVA) is a widely utilized encapsulating material due to its generally-recognized-as-safe (GRAS) status, high oxygen barrier property, and water solubility, which facilitates the easy recovery of bacteria [152,153]. PVA can also be combined with other materials to improve it. Xuejun Kang et al. found that the addition of silk fibroin (SF) to PVA can solve the issue of pure PVA films being too thin [154]. Hou Juncai et al. successfully prepared core-shell fibers encapsulating Lactobacillus paracasei using coaxial electrospinning technology and Eudragit S100 (ES100) and PVA/pectin (PEC). The results indicated that the fibers could enhance the tolerance of probiotics to the adverse environment in the gastrointestinal tract. The prepared fibers could be used to prepare functional fermented foods in the future [155]. Enes Dertli et al. developed a new electrospun nanofiber of poly(vinyl alcohol)/sodium alginate (PVA/SA) blends as an encapsulation material to extend its application in the packaging of living organisms, such as probiotics. They found that the PVA/SA blends exhibited biocompatibility, nontoxicity, and good chemical and thermal stability [156].

Table 3 presents a compilation of published literature on the encapsulated microorganisms, electrospun materials and parameters, and average nanofiber diameter. Generally, unloaded nanofiber mats exhibited a consistent, smooth, and beadless morphology [157]. SEM images from the literature revealed that encapsulation of probiotics resulted in a “string and beads” or spindle-fiber morphology, attributed to the size of probiotics that exceeded the fiber diameter [158]. The concentration, flow rate, voltage, and distance of the solution in the electrospinning process need to be experimentally validated to obtain the optimal parameters: (1) Concentration of solution: When the concentration of the solution is very low, polymeric micro (nano) particles will be obtained [159]. Due to the low viscosity and high surface tension of the solution, electrospray rather than electrospinning will occur at this time [160]. When the concentration is slightly higher, a mixture of beads and fibers will be obtained. When the concentration is appropriate, smooth nanofibers can be obtained. If the concentration is too high, spiral microbands can be observed. (2) Voltage: Some researchers believe that a higher voltage is conducive to the formation of larger fiber diameters. A voltage that is too high will result in beads and cause the solution to be rapidly removed from the needle tip. A lower voltage helps to produce uniform nanofibers with a narrower diameter distribution [161,162]. Wael Mamdouh et al. compared the nanofibers formed under 16–20 kV voltage and found that the fiber distribution was uniform under the 16 kV condition, which was the optimal voltage [163]. (3) Flow rate of the polymer solution: When the flow rate is high, beads with larger diameters will be formed due to the short drying time and low stretching force [164]. (4) Distance between the collector and the tip of the injector: If the distance is too short, the fibers will not have enough time to solidify before reaching the collector, while a longer distance will result in thinner fibers. However, if the distance is too long, bead-shaped fibers can be obtained [165,166]. It is worth noting that we found that overall fibers produced by blended spinning cannot protect probiotics from damage under the acidic conditions of the stomach, while coaxial electrospinning can provide better protection. The use of nanofiber mats encapsulating Lactobacillus casei in PVA/polyethylene oxide copolymer (PEC) was analyzed for its heat stability and simulated digestion [155]. This strategy has higher thermal stability and can protect probiotics from gastrointestinal damage, as well as improve their adherence and growth in the intestine. In general, coaxial or multilayer electrospinning is more effective in protecting the incorporated probiotics [167].

**Table 3 nanomaterials-13-02185-t003:** Different electrostatic spinning methods employed for the encapsulation of probiotic microorganisms.

Reference	Bacteria Systems	Polymers	Processing Parameter	Nanofiber Average Diameter
Hong Wu. et al. [168]	*Lactobacillus plantarum*	PVA	16 Kv, 0.3–0.6 mL/h, 14 cm	410 ± 150 nm
Enes Dertli et al. [156]	*Lactobacillus paracasei* KS-199	PVA, sodium alginate (SA)	22 Kv, 1.2 mL/h, 10 cm	305 nm
Fauzia Yusuf Hafeez. et al. [169]	*Enterococcus mundtii* QAUEM2808	PVA, PVP, glycerol	16 Kv, 0.6 mL/h, 15 cm	318 nm
Juncai Hou. et al. [170]	*Lactobacillus encapsulation*	gum arabic (GA), pullulan (PUL)	16 Kv, 0.4 mL/h, 10 cm	105–283 nm
Wael Mamdouh. et al. [163]	*Lactobacillus*	PVA, inulin	16 Kv, 0.6 mL/h, 10 cm	200–400 nm
Anja Boisen et al. [171]	*LGG*	Pullulan, Poly-lactic-co-glycolic acid (PLGA)	12 Kv, 1 mL/h, 15 cm	287 ± 102 nm
Huda Ateeq. et al. [172]	*Lactobacillus acidophilus*	Gum Arabic (GA) and PVA	16.8 Kv, 90 mm/s, 15 cm	617 nm
Adem Gharsallaoui. et al. [173]	*lactobacilli*	chitosan (CS), PVA	18 Kv, 0.1 mL/h, 15 cm	117.5 ± 70.6–217.6 ± 62.7 nm
Maryam Azizkhani. et al. [174]	*Lactobacillus* and *Bifidobacterium*	corn starch (CS), SA	24 Kv, 1.5 mL/h, 12 cm	295 nm
Bin Jiang. et al. [175]	*Lactobacillus plantarum*	polylactic acid (PLA)	16 Kv, 0.25 mL/h, 15 cm	676 ± 162 nm
Zsombor K. Nagy. et al. [176]	*Lactobacillus*	PVA, polyethylene oxide (PEO)	40 Kv, 20 mL/h, 35 cm	100 nm
Min-Tze Liong. et al. [177]	*Lactobacillus acidophilus*	PVA, soluble dietary fiber (SDF),	12 Kv, 0.1 mL/h, 15 cm	229–730 nm

Nevertheless, electrospinning technology faces the challenge of accommodating the development of delivered probiotics. The properties of probiotics, the compatibility between probiotics and matrix materials, and the uncertain correlation between electrospinning parameters and optimal probiotic release profiles are all relevant factors. The complexity of these factors makes it difficult to electrospin probiotic-loaded fibrous membranes efficiently and on schedule [178,179]. Furthermore, encapsulating probiotics in nanofibers is aimed at protecting their viability during processing, storage, and consumption. Therefore, the impact of the electrospinning process on probiotic viability has been a prominent area of research. The decline in probiotic viability following electrospinning is likely caused by a significant alteration in the osmotic environment resulting from the swift evaporation of water during the process [180,181]. Although required for nanofiber production, the high voltage utilized in electrospinning may be detrimental to probiotics. During the process of electrospinning, certain additives within the solution, such as edible fat [182], can serve as an effective protective layer around the enclosed probiotics, ensuring their viability.

#### 4.1.3. Other Techniques

Self-assembly is a remarkable design principle found in nature, whereby molecular components arrange themselves spontaneously into hierarchical structures. Many complex biological structures, such as proteins, viruses, and cell membranes, are formed through dynamic self-assembly, which involves a series of energy-consuming assembly and disassembly steps. These processes intricately regulate the aggregation of biomolecules to create various cellular components, such as filaments, membranes, and organelles, that execute a diverse array of biochemical reactions essential for sustaining. Zhenzhong Zhang et al. designed a multifunctional self-assembling coating to encapsulate a super gut microbe (SGM) targeted at *E. coli* Nissle 1917 to enhance intestinal colonization. In a mouse model of Salmonella enterica serovar Typhimurium (STm) colitis, SGM treatment resulted in a 6.8-fold reduction in STm compared with untreated probiotics [183]. The system increased the survival rate of probiotics under acidic and bile conditions and improved their adhesion to intestinal mucosa compared to untreated probiotics. Moreover, the system could autonomously regulate the pathological microenvironment (such as scavenging inflammation-mediated ROS and removing iron from pathogenic bacteria to further improve the probiotic survival rate), synergistically enhancing their colonization in diseased intestines. In another study, Md. Arifur Rahim et al. used the metabolic regulation mechanism of probiotics on manganese metal to achieve the self-assembly of polyphenols on the surface of lactic acid bacteria (LAB) cell walls [21]. This encapsulation endowed the cells with additional functions, such as tissue adhesion and antioxidant activity. This modification method has great potential for horizontal expansion, as the polymerization reaction can be catalyzed by a variety of transition metals, and the rich metabolic pathways of bacteria and cells to metals provide a variety of possibilities for expanding the application of this method. Jia Lingyun et al. used a bacteria-induced polyphenol colloid particle aggregation technique to enhance the oral bioavailability of probiotics. Within 10 s, 97% of the bacteria were rapidly encapsulated in the colloidal shell, with a raw material utilization rate of 91%. In vitro simulated experiments showed that the tightly packed, thick, and positively charged colloidal shell effectively protected the probiotics from simulated gastric acid erosion, with a survival rate of 19%. This is 7500 times higher than commercial enteric-coated material L100. The oral availability of encapsulated probiotics increased by about five times compared to non-encapsulated probiotics [184].

Three-dimensional printing is praised as a disruptive technology that will change the manufacturing industry [185]. It is used in various fields, such as aerospace, defense, art and design, and food. Three-dimensional printing allows for structures with high surface-to-volume ratios [186] and has demonstrated tremendous commercial potential due to its advantages in the customization of food and food packaging materials, its multifunctionality, and its ability to create complex designs [187]. In recent years, researchers have attempted to combine 3D printing with probiotic encapsulation, hoping to open up new paths for the preparation of functional probiotic foods. Yimin Lou et al. evaluated the printability of dough formulations with different water contents, wheat flour types, and calcium caseinate usage, and found that the geometric shape of the 3D printed structure containing probiotics was well preserved during the baking process [188]. Therefore, the application of 3D printing in probiotic encapsulation is feasible. Notably, the nozzle diameter, feed rate, shear stress, and shear rate during 3D printing can significantly affect the activity of probiotics. Min Zhang et al. used fully gelatinized potato chip starch as the raw material, optimized the control conditions of the 3D printing process, and prepared instant mashed potatoes enriched with probiotics [189]. This method can avoid high temperatures during the cooking process and better maintain the activity of probiotics. Based on the unique advantages of 3D printing, the ability to create internal structures with different material densities and design complexity provides a new path for the development of functional probiotic foods. The controllable surface-to-volume ratio of 3D printing creates enormous potential for customization of probiotic foods, and we can expect to see more forms of functional probiotic foods in the future.

The above methods can be combined to prepare encapsulated probiotics, as shown in Figure 3.

### 4.2. Methodologies for Testing the GI Behavior of Encapsulated Bacteria

The digestion process can be studied using both in vivo (human or animal) and in vitro methods, each with its own sets of advantages and disadvantages. While in vivo studies may offer direct and personalized results, they can be expensive and require ethical considerations [190]. In contrast, in vitro models are generally preferred for their cost efficiency, reproducibility, and standardized conditions [191]. However, they may not fully represent the complexity of the human digestive system and may not account for various individual differences and factors. Therefore, both approaches can be useful in food, nutrition, and medical research.

#### 4.2.1. Criteria for Assessing the Quality of Probiotic Encapsulation

The size of the carrier particles directly impacts various factors such as bio-distribution, stability, and absorption efficiency. The optimal range for carrier particles is recommended to be 100–200 µm, with the boundary value for the differentiation between micro- and nanoparticles, being 100 nm [192]. Microparticles provide sufficient area and capacity for probiotic encapsulation, while nanoparticles offer a promising delivery system with enhanced bioavailability [193]. The core-to-wall ratio and encapsulation process also contribute to the final particle size [194]. Overall, particle size plays a critical role in determining the effectiveness of encapsulated probiotics.

Viability refers to the quantity of encapsulated probiotic cells (expressed as cfu g^−1^) that remain capable of producing a favorable health effect in the host’s site of action [14]. High temperatures can damage the cell membrane, denature proteins, and lead to the death of probiotics [195]. Even relatively lower temperatures can reduce the viability of probiotics [196]. Both excessively high and excessively low water activity can lead to cell damage and protein denaturation. The optimal range for water activity is 0.25, with a moisture content of 4–7% [197,198]. High water activity can increase bacterial mortality, while low water activity can affect cell oxidation [199]. Increasing pressure to 50 MPa disrupts the cell division and protein synthesis rate of *E. coli* [200]. However, research has also shown that pre-treatment with pressure can increase the thermal tolerance of microorganisms [201]. Oxygen toxicity is a significant issue for anaerobic microorganisms. When packaging anaerobic cultures, it is recommended to include a strain with high oxygen consumption, such as Streptococcus thermophilus [202]. This effectively reduces the oxygen, which can be hazardous to anaerobic cultures. A completely anaerobic environment should be established during the packaging process, including anaerobic sealing solutions, to maintain an oxygen-free environment for the packaging tools. Additionally, substances such as L-cysteine or ascorbic acid can be added to lower the redox potential to facilitate the growth of anaerobic organisms [203,204].

Elasticity refers to a material’s ability to resist deformation and recover its original structure [205]. Hydrogels have adjustable elasticity due to their hydrophilic and porous properties, which can be controlled by varying their internal structure and cross-linking density. The cross-linking method and choice of crosslinkers significantly affect the elasticity of hydrogels [206,207]. Natural crosslinking agents such as genipin, phytic acid, and transglutaminase that have been used to stabilize microgels and enhance probiotic protection [208,209]. The elasticity of layer-by-layer (LbL) or templated capsules can also be adjusted by altering the materials, number of layers, thickness, and crosslinking. Despite advancements in understanding hydrogel elasticity, its role in particle-active delivery systems in the food sector has been relatively underexplored.

Zeta potential refers to the electrokinetic potential in colloidal systems, and it influences the stability and behavior of encapsulated products in the digestive tract [210]. The amplitude of zeta potential, which ranges from −200 to +200 mV, impacts the stability of particles against agglomeration or coagulation. Factors such as composition, concentration, pH, ionic strength, and additives affect the zeta potential. The modulation of the gelling process in hydrogel delivery systems, such as whey protein-based gels, also depends on regulating the zeta potential [211,212]. However, challenges remain in accurately measuring zeta potential in hydrogel matrices and considering other factors like viscosity and intramolecular interactions [213]. Particle size also plays a crucial role in zeta potential measurements for micro- and nanocapsules [214].

Encapsulation efficiency (EE) measures the extent to which a bioactive ingredient is encapsulated within an inert core material [215]. A high EE indicates effective encapsulation, which protects the bioactive ingredient from degradation and maximizes its stability [216]. Various variables, such as the carrier material and purification degree, can influence EE [217]. For example, using pure phosphatidylcholine as a carrier material can achieve high EE [218]. Optimizing factors like core-wall ratio, homogenization process, and encapsulation technology can maximize EE and improve stability [219,220,221]. EE is also crucial in manufacturing commercial supplements and probiotic products. Encapsulation using polysaccharide and protein carriers generally yields higher EE than lipid-based encapsulation [216]. Methods involving strong shear forces, high pressure, and high temperature tend to result in higher EE, while milder methods provide lower EE [216].

#### 4.2.2. In Vitro Methodologies

In vitro testing is often used to evaluate novel microencapsulation systems in their early stages [222]. Microgel architecture is typically characterized using optical or atomic force microscopy, providing valuable information about the microgel surface morphology [223]. When detecting the structure and distribution of probiotics in microgels, fluorescence dyes, such as FITC, can be selectively labeled, and their positions can be visualized using fluorescence microscopy. Fluorescent dyes are also used to measure the internal pH of microgels and the location and viability of probiotics [224,225]. The Live/Dead Backlight Bacterial Viability Kit can be used to detect the activity of probiotics. After staining, qualitative images can be obtained using confocal microscopy, or the proportion of living and dead cells can be quantitatively detected using flow cytometry [226,227]. The antibacterial zone assay can be used to investigate the inhibitory effect of probiotics on harmful bacteria, and *E. coli* and *Staphylococcus aureus* are often used as competitors [228]. To evaluate the efficiency of probiotic delivery systems, the tolerance of probiotics during gastrointestinal transport should be assessed. The survival ability of probiotics can be determined using plate counting or flow cytometry, as described by the International Organization for Standardization (ISO, 2015) [229,230].

Since the 1990s, in vitro digestion models have been developed and utilized in food digestion studies [231]. These models serve as tools that can aid in the conscious design of food products for human health by estimating the in vivo behavior after meals. There are two types of in vitro digestion methods for evaluating probiotic survival during transportation: static and dynamic. The static model uses a constant ratio of food to fluid, with simulated saliva, simulated gastric fluid (SGF), and simulated intestinal fluid [232]. In addition, static models are a practical, affordable, and viable option to evaluate numerous experimental conditions and a vast array of samples [233]. The current literature primarily employs basic models of the human gastrointestinal (GI) tract, making it challenging to compare results due to inconsistent compositions and varying pH levels of simulated GI solutions [234,235]. The most frequently used simulated gastric solutions are based on the United States Pharmacopeia recipe, which utilizes solutions of HCl and salts at a pH of approximately 1.0–2.0 [236,237]. A more precise simulation of the actual digestive process can be achieved using a dynamic model, which can regulate the pH, control the flow of food, and inject digestive enzymes in real time at different compartments of the gastrointestinal tract (GIT) [238]. Marteau et al. have successfully validated an in vitro multicompartmental simulation of the stomach and small intestine to test the viability of lactic acid bacteria during digestion. The simulation divides the model into four compartments that imitate the stomach, duodenum, jejunum, and ileum, as shown in Figure 4 [239]. The model provides a complex chemical environment for the study of probiotics, but adding a bacterial strain to the highly diverse gut microbiota will have a greater impact on the microbial community within the gut, beyond simply counting live cells. To further investigate this, Molly et al. designed the Simulator of Human Intestinal Microbial Ecosystem (SHIME) [240]. The system initially consisted of five vessels with complex culture media inoculated with fecal material simulating the duodenum, jejunum, ileum, ascending colon, transverse colon, and descending colon. De Boever et al. improved the SHIME system by adding a vessel simulating stomach activity to enhance its simulation accuracy of the entire gut environment [241]. Table 4 summarizes the advantages and disadvantages of various in vitro dynamic digestion models.

**Table 4 nanomaterials-13-02185-t004:** Traits of in vitro dynamic models utilized in digestion research.

Reference	System	Type	Advantage	Disadvantage
Mans Minekus. et al. [242]	The TNO gastro-intestinal model (TIM)	Multi-compartmental system	Reliable and cost-effective in vitro tool which fully assimilates the gastrointestinal tract, mimics crucial parameters of human digestion.	Lacks feedback on the GI conditions of the energy density of the food and is limited in simulating the anatomy of each digestive phase, as well as measuring bioavailability rather than bioaccessibility of a compound.
Molly Koen et al. [243]	The simulator of the human intestinal microbial ecosystem (SHIME)	Multi-compartmental system	Can maintain the stability and interaction of microbiota for a long time, and accurately evaluate the effects of drugs and food treatments. It can also study inter-individual variability of microbial communities.	Lacks a realistic physiological environment and anatomy in the digestive tract.
Guerra, Aurélie. et al. [244]	The new engineered stomach and small intestine model (ESIN)	Multi-compartmental system	Broad range of applications but needs further validation for applications outside of liquid drug digestion. In addition, can simulate the sieving effect of the gastric pylorus and allow the real-size food bolus to enter the stomach.	Cannot mimic the anatomy of each digestive organ nor simulate colonic fermentation. The model also struggles to simulate peristalsis and contractions in small intestinal chambers.
Daniel Picque. et al. [245]	The dynamic gastrointestinal digester (DIDGI)	Multi-compartmental system	Able to evaluate the digestion of infant formula and cheese-ripening microbiota in the GI tract and flexible working environment and can hold most masticated food. In addition, transparent chambers for direct observation of biochemical and physical processes during digestion.	Lacks the capability to simulate essential biomechanical and anatomical aspects of food digestion as well as the signals that control digestion speed and satiety.
Barroso, Elvira. et al. [246]	The simulator of the gastrointestinal tract (SIMGI)	Multi-compartmental system	Can reproduce stable microbial communities and explore the effects of diet and food on microbiota. It has automatic control and flexibility.	Lacks the ability to simulate gut microbiota–host interactions, metabolite absorption, and digestive anatomy.
Norwich et al. [247]	Dynamic gastric model (DGM)	mono-compartmental system	Simulate real human gastric biochemical conditions, predict the influence of food on health, and test the solubility of orally administered solid drugs.	Cannot provide visual observation, is exposed to air, and can only simulate a part of the gastric digestion process.
Fanbin Kong. et al. [248]	The human gastric simulator (HGS)	mono-compartmental system	Simulate the human digestive process, combine secretion, emptying, and temperature control, and hold several liters of material for digestion studies. Additionally, the solid-state polyester mesh bag in HGS allows for particle screening, simulating the sieving effect of the pylorus.	Only simulates stomach digestion and is costly for testing high-value ingredients due to sample size limitations. In addition, not transparent, and does not provide feedback on pH control, limiting its ability to assess processed foods in real-time.
Kozu Hiroyuki et al. [249]	The gastric digestion simulator (GDS)	mono-compartmental system	Allow for quantitative analysis and real-time visualization of the digestion process, as well as simulating peristaltic motion and mixing	Need for validation against in vivo data and does not mimic the characteristic “J” shape of human gastric morphology. Additionally, the number and location of the rollers used in the models are not fully adequate in simulating the real stomach peristalsis, and they only simulate one compartment of the human digestive tract.
Awad, T. S. et al. [250]	The in vitro mechanical gastric system (IMGS)	mono-compartmental system	It simulates the shape and peristaltic waves of the human stomach and can adjust the pressure and monitor the pH value. Compared with other models, gastric peristalsis is more realistic.	It does not simulate the filtering function and continuous secretion of gastric juice, which may affect the accuracy of simulation results. More experiments are needed to verify its practicality and reliability, and its ability to handle gastric emptying is relatively simple.
Cordonnier, Charlotte. et al. [251]	The artificial colon model (ARCOL)	mono-compartmental system	Semi-continuous fermentation system with various ports and probes, making it suitable for studying the survival rate of yeast probiotics and their effects on intestinal microbial metabolism. The system is capable of maintaining anaerobic conditions through the activity of gut microbiota and has a hollow fiber membrane to simulate passive absorption of metabolites.	Does not distinguish between different colon conditions and lacks mucosal contact surface.

#### 4.2.3. In Vivo Methodologies

Animal experiments can also provide certain research data to support and guide the conduct of clinical trials [252]. By conducting preliminary experiments on animals, scientists can better understand the safety and efficacy of the research object, which helps to reduce risks in human populations and ensure the effectiveness and sustainability of treatments. The morphology of the stomach and emptying characteristics of dogs are comparable to those of humans, whereas the colon morphology of pigs appears similar to that of humans [253]. Based on this judgment, we can conduct animal experiments to guide subsequent clinical trials. Microencapsulated probiotic cells can be studied through real-time PCR and fluorescence in situ hybridization (FISH) after oral administration to experimental animals [254]. Ma Yuan et al. used dogs as experimental animals to explore the protective effect of microencapsulation on *Lactobacillus acidophilus* [255]. Tzortzis et al. studied the effect of a “prebiotic” using pigs [256].

Although the gastrointestinal tracts of rodents, especially rats and mice, are not very similar to those of humans, they are often used to test probiotics due to their ease of breeding and more economical characteristics. Jia Lingyun et al. used positively charged colloids (NTc) composed of amino-modified poly-β-cyclodextrin and tannic acid to encapsulate negatively charged probiotics, showing strong resistance to simulated gastric acid with a survival rate of up to 19%, which is 7500 times higher than that of the commercial enteric material L100. The encapsulated probiotics showed good therapeutic effects in a colitis mouse model [184]. Chen Qian et al. utilized electrostatic self-assembly to encapsulate *E. coli* strains expressing catalase and superoxide dismutase with chitosan and sodium alginate as effective biofilms. In different chemically induced mouse models of inflammatory bowel disease (IBD), the results were characterized by recording intestinal photos and integrity and performing histological staining and immunofluorescence staining of the intestine. This composite was able to effectively alleviate inflammation, repair colonic epithelial barriers, regulate intestinal microbiota, and increase the abundance of important microbial species that maintain intestinal homeostasis in the gut microbiome [257]. In further research, more convincing data can be obtained by using animals such as pigs or dogs, which are more similar to humans, although there are higher costs and more ethical issues involved.

## 5. Challenge of Commercializing Encapsulated Probiotics

The development and screening of new food-grade and pharmaceutical-grade biocompatible wall materials are essential for research into probiotic nanoformulations. These materials should ensure the survival of probiotics in harsh conditions and secure the integrity of probiotic nanoformulations in vivo. However, due to individual variability in bodily environments, particularly among patients with considerable physiological differences, there are significant challenges. Despite advancements in methods such as material wrapping, modification, or encapsulation to deliver probiotics, these techniques often provide only temporary protection against physiological gastrointestinal conditions. Unfortunately, they often prove ineffective in pathological conditions such as inflammatory bowel disease (IBD). The task of designing universally applicable probiotic preparations targeting various gut microenvironments remains a formidable challenge. Researchers can encapsulate probiotics by developing materials that enable site-specific release in pathological environments. For example, Haibo Mu et al. designed a ROS-responsive hydrogel based on hyaluronic acid (HA). This hydrogel selectively cleaves disulfide bonds in response to excess reactive oxygen species (ROS) produced in inflamed colon tissue, leading to hydrogel degradation and localized probiotic release [258]. A range of encapsulation techniques have been developed that prolong the lifespan of probiotics and facilitate the commercialization of various encapsulated probiotic products. However, obstacles remain on the path to commercialization. Despite promising laboratory-scale results, difficulties emerge when scaling up the technology for industrial applications. For instance, in the case of the extrusion method, low production capacity and large particle sizes must be taken into account [259]. Additionally, stability issues arise during probiotic storage. Most probiotic products require refrigeration, and little research suggests an appropriate storage temperature for encapsulated probiotics [122,260,261]. In general, the viability of probiotic bacteria during storage is inversely related to the storage temperature. It is recommended to store probiotic food products at a temperature of 4–5 °C [262]. However, different probiotic strains may have different optimal storage temperatures, and the optimal storage temperature may also vary depending on the packaging technology used. For example, *Bifidobacterium lactis* BB-12 has an optimum storage temperature of 8 °C [263], while freeze-dried probiotics are best stored at −18 °C [264]. Therefore, it is advisable for researchers to determine the optimal storage temperature for their specific product during the research process. Moreover, after opening the packaging, probiotics should also be refrigerated since high atmospheric humidity can lead to their deactivation or degradation [265]. Foods with high organic acid content, such as yogurt or juice, are even less favorable for probiotic survival under acidic conditions [266]. During commercial operations, the production, transportation, storage, and sales processes should be conducted under low-temperature conditions to enhance the product’s shelf life.

## 6. Conclusions and Future Scope

This article explores the significance of probiotics, the importance of encapsulation, the materials and techniques employed for encapsulation, and the methods of confirming the effectiveness of encapsulation. Probiotics have a crucial role in addressing diverse health issues, such as enhancing gastrointestinal health, strengthening the immune system, supporting metabolic regulation, and influencing mood. Studies suggest that probiotics can also help prevent viral infections through the gut-lung connection [267,268]. This suggests the potential use of probiotics in managing respiratory illnesses caused by viruses, including those experienced during the COVID-19 pandemic in 2020, as well as neurological conditions like influenza virus infection [268,269]. The encapsulation of probiotics has demonstrated potential in safeguarding these microorganisms and improving their precise delivery [270]. As our knowledge of probiotics deepens, their crucial role in human health and potential for addressing various diseases become increasingly evident. Encapsulation technology holds significant value for probiotics, providing researchers and industry professionals in this field with the latest tools to advance this technology from the laboratory to industrial applications. It is important to note that although numerous research studies are published in this area, only a few commercially available products incorporate encapsulated probiotics. Manufacturers must consider factors such as cell viability and probiotic functionality in order to make legitimate health claims. Therefore, laboratories face the challenge of developing viable technologies for industrial production while maintaining appropriate scale and cost control. In conclusion, the prospects for the development of the probiotics market are immense, and laboratory research holds the potential to transition towards industrialization.

## Figures and Tables

**Figure 1 nanomaterials-13-02185-f001:**
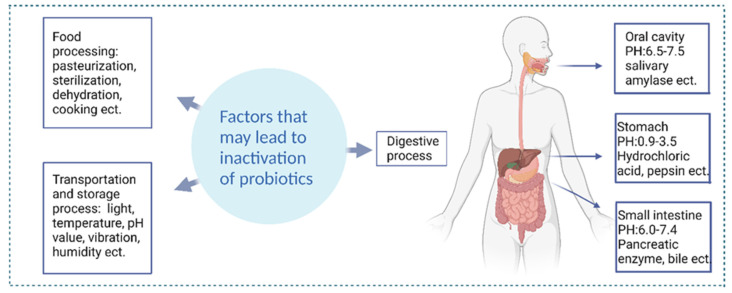
Factors that can lead to the inactivation of probiotics.

**Figure 2 nanomaterials-13-02185-f002:**
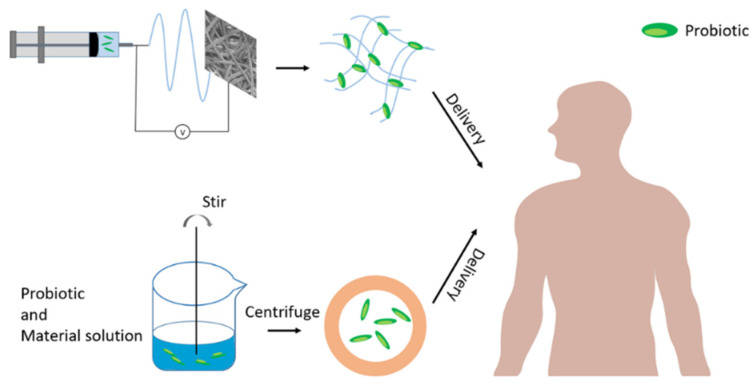
Microencapsulation and electrospinning for encapsulation of probiotics.

**Figure 3 nanomaterials-13-02185-f003:**
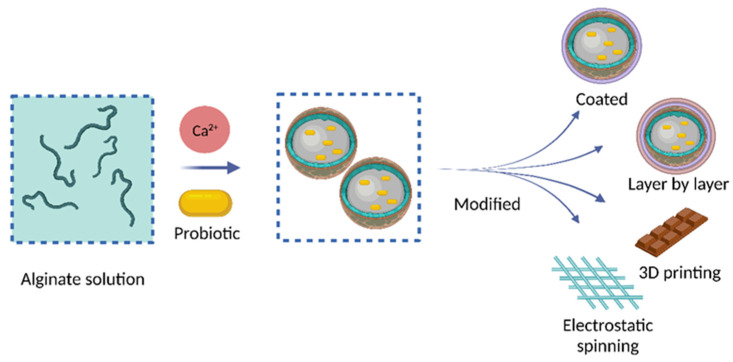
A combination of methods to encapsulate probiotics.

**Figure 4 nanomaterials-13-02185-f004:**
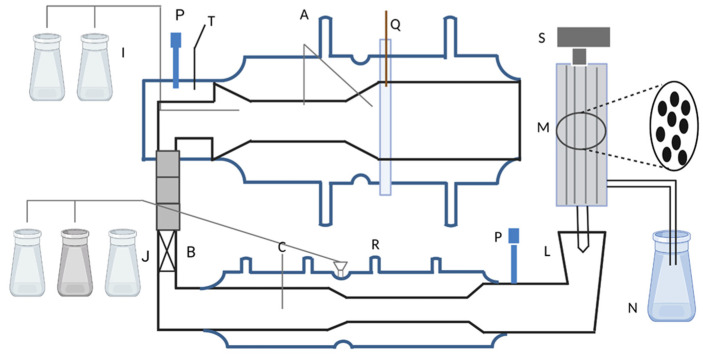
The TNO gastro-intestinal model: A: Stomach; B: Peristaltic valve (controlling gastric emptying); C: Small intestine; I: Simulated gastric secretion; J: Duodenal secretion; L: Pre-filter; M: Hollow fiber semi-permeable membrane: N: Filtrate collection for analyzing bioaccessible API from the small intestine (N); P: H electrode; Q: pressure sensor; R: temperature sensor (in eachcompartment); S: liquid level sensor; T: inlet for the meal.

## Data Availability

No new data were created or analyzed in this study. Data sharing is not applicable to this article.

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
