# Peer review of "Biomaterials and Encapsulation Techniques for Probiotics: Current Status and Future Prospects in Biomedical Applications"

_nanomaterials, 2023, doi:10.3390/nano13152185_

Round 1

Reviewer 1 Report

I consider the review to be complete, interesting and well organized and therefore suitable for the journal's issue.

  I consider the present review relevant in the field of biomaterials applications. Figures and tables are intuitive. References are appropriate and recent. In my opinion, the conclusion section could be improved in order to resume presented data and better discuss future prospects.

Author Response

Thank you for your recognition and suggestions. The conclusion part has been revised.

Reviewer 2 Report

The manuscript entitled "Biomaterials and Encapsulation Techniques for Probiotics: Current Status and Future Prospects in Biomedical Applications" could be interesting but some improvements are required before being considered for publication:

Some minor spelling mistakes have been detected. Please revise and correct.

Introduction

It is required to rewrite this section because too many repetitions have been detected.

It is required to highlight the novelty of this review since many reviews have been published dealing with the encapsulation of probiotics.

What criteria have the authors followed to select the works reflected in this review?

Section 2

This section also needs to be reorganized to improve readability and avoid repetition.

Section 3

Why did the authors focus on polysaccharides? Which other biomaterials have been used for encapsulating probiotics? Multiple matrixes have been used so far.

Section 4

Authors differentiate between microencapsulation and electrospinning. In fact, electrospinning is a microencapsulation technique. However, it does not produce spherical particles but fibers.

Regarding electrospinning as an encapsulation technique, it is required to take into account the impact of the fibers in the final product. Since it can impact the texture of a food product.

In my opinion, emulsions are delivery systems but not properly encapsulation systems. In table 2, authors mention particle size, but in case of emulsions the correct terms is droplet size.

What do authors mean when they say “The complexity of these factors makes it difficult to electrospin probiotic-loaded fibrous membranes efficiently and on schedule”?

It is required to provide also discussion about the advantages and disadvantages of the different encapsulation technologies regarding probiotic encapsulation. As well as the possibility to reproduce lab results at industrial scale.

When encapsulated probiotics are added to a food product, the capsules must resist not only the food production and storage but also digestion. Many of the scientific works revise one aspect or the other but not both. It would be required to highlight this aspect in the manuscript.

Section 5

Authors mention that there are still obstacles that prevent encapsulated probiotics commercialization. More emphasis and discussion is required dealing with these obstacles.

Some minor spelling mistakes have been detected. Please revise and correct.

Author Response

Introduction: 

Thank you for the comment. The introduction section has been rewritten.
"This review aims to elucidate and integrate the advantages and disadvantages of various encapsulation techniques, materials, evaluation methods, etc., for probiotics, with the goal of promoting the advancement in this field. " This sentence emphasizes the innovation of this article and has been added to the introduction section.

In this review, the criteria for selecting the articles as examples are as follows:
Academic influence: Prioritizing works that have high academic reputation and impact in the field.

Alignment with previously mentioned research methods and design.
Time range: Giving preference to recent works to reflect the latest research advancements and technological developments.

Section 2

Thank you for the comment. This section has been revised.

Section 3

Nature biomaterials, protectants, prebiotics, and synthetic materials are used for encapsulation. In natural materials, polysaccharides such as alginate and chitosan are among the most commonly used. Protein-based materials like gelatin are also frequently used as encapsulating agents. In summary, the materials discussed in this article are selected based on their wide adoption by researchers. In addition to the mentioned materials, starch, xanthan gum, pectin, and others are also commonly used for encapsulating probiotics.

Section 4

Some scholars consider electrospinning as a form of encapsulation, while others believe that traditional microencapsulation does not include electrospinning. In this case, due to the fact that electrospinning produces fibers, and commonly used materials are polymer materials such as PCL and PVA, a distinction is made.

Emulsion is a form of microencapsulation. The terminology has been adjusted, thank you for the clarification.

“The complexity of these factors makes it difficult to electrospin probiotic-loaded fibrous membranes efficiently and on schedule”: The properties of probiotics, the compatibility between probiotics and matrix materials, and the uncertain correlation between electrospinning parameters and optimal probiotic release profiles are all relevant factors. The complexity of these factors makes it difficult to efficiently and on schedule electrospin probiotic-loaded fibrous membranes.

The advantages and disadvantages of the Feature section have been added, and the possibility to reproduce lab results at an industrial scale has been included. Thank you for the suggestion.

"When encapsulated probiotics are added to a food product, the capsules must resist not only the food production and storage but also digestion. Many of the scientific works revise one aspect or the other but not both. "This section has been added in section 4.1. Thank you.

Section 5

The suggestions, such as considering specific packaging materials under pathological conditions and recommended storage temperature for probiotics, have been incorporated into the translation.

Reviewer 3 Report

The review describes the importance of probiotics, the need for encapsulation, the materials and methods used. The topic is currently very interesting and the authors have discussed the importance of probiotics and the role that they play in human health. Some prospects on industrial market are highlighted in order to make the transition toward industrialization.

 However, I suggest aspects should be investigated by the authors before publication:.

-In the section 3.“Polymer choice….” (line 125) a more general perspective on the chemical nature of hydrogel materials employed for the delivery of probiotics is missing. The authors should mention the use of hydrogels other than those based on polysaccharides and also mention those based on protein(DOI 10.1016/j.colsurfb.2021.111989)

- Moreover a crucial aspect for hydrogels applications is their mechanical properties: their evaluation is necessary as reported from other authors (DOI10.1088/1361-6528/abc5f6)

- At page 5 section 3.2, chitosan is proposed for probiotic encapsulation. The authors mentioned ionotropic gelation as a method to obtain chitosan gels. I suggest to include also freeze gelation as a common employed method and cite proper references for both ionotropic gelation (https://doi.org/10.1016/j.molstruc.2021.132129) and freeze gelation (https://doi.org/10.1016/j.carbpol.2021.118156)

- Finally a general English revision is needed

Author Response

  1. Thank you for the suggestion. It has been modified. In Section 3, the materials are categorized as polysaccharides, proteins, lipids, and synthetic polymers.
  2. Thank you for the suggestion. Section 4.2.1 has been added.
  3. Thank you for the suggestion. This section has already been added in section 3.1, line 201.
  4. I will review the language of the entire text again.

Reviewer 4 Report

The review "Biomaterials and Encapsulation Techniques for Probiotics: Current Status and Future Prospects in Biomedical Applications" is an interesting and high-quality work on a topical topic.

I would strongly recommend that the authors of the review article make a schematic drawing that reflects the trends in the use of Biomaterials and Encapsulation Techniques for Probiotics. Drawings like this always tame reviews and boost citations.

In my opinion, section 3 lacks a presentation of the chemical structure of the described polymers. This is an important factor in their suitability and effectiveness for the immobilization of microorganisms.

In general, the article lacks graphic material. Additional illustrations of Probiotics encapsulation approaches from the cited papers would certainly enhance the review article.

In my opinion, the article is not quite in line with the profile of the Nanomaterials magazine. There is no clear emphasis on the use of nanomaterials in the work. However, this remark is at the discretion of the editor-in-chief of the journal.

Author Response

Section 3 has already added chemical structural formulas, thank you for the suggestion.

Schematic drawing has already added.

Thank you once again for your comments.

Round 2

Reviewer 2 Report

The manuscript entitled "Biomaterials and Encapsulation Techniques for Probiotics: Current Status and Future Prospects in Biomedical Applications" has been improved but some improvements are still required before being considered for publication:

Authors mention “Iolanda Francolini et al. developed chitosan scaffolds with improved mechanical properties using methods such as thermally induced phase separation, cryogelation, and photocrosslinking. The pore size of the scaffolds was suitable for osteoblast cell growth, and the compressive modulus of the scaffolds increased by approximately three times. Furthermore, the crosslinked scaffolds exhibited good elastic recovery without the need for toxic cross linking agents. This method can be extended to other soft materials [67]”, I do not see the relationship between scaffolds for osteoblasts and probiotic encapsulation. Please explain it or remove it.

If probiotics are in the micron size range, how can nanoparticles contribute to this application?

Regarding the factors that affect probiotic encapsulation, probiotic viability after the encapsulation process is one of the main parameters and it has not been considered for the authors.

Grammatical errors have been detected. Please revise and correct.

Author Response

It has been modified, thanks for your guidance.

Indeed, probiotics are indeed micro-sized, and the process of microencapsulation allows nanoparticles to serve as carriers for probiotics, effectively delivering them to the target location. Nanoparticles provide a protective barrier, and they also have controlled release and sustained release capabilities. Moreover, nanoparticles can be surface-modified and functionalized to enhance the effects of probiotics.

"Viability" has been added to the review, thanks for the guidance